# Navigating agricultural nonpoint source pollution governance: A social network analysis of best management practices in central Pennsylvania

Elsa L. Dingkuhn[1,2]*, Lilian O'Sullivan[2], Rogier P. O. Schulte[1], Caitlin A. Grady[3¤]

1 Farming System Ecology group, Wageningen University and Research, Wageningen, the Netherlands, 2 Crops, Environment and Land Use Programme, Teagasc, Wexford, Ireland, 3 Department of Civil and Environmental Engineering, Rock Ethics Institute, Pennsylvania State University, University Park, PA, United States of America

¤ Current address: Department of Engineering Management and Systems Engineering, George Washington University, Washington, DC, United States of America
* elsa.dingkuhn@wur.nl

**Data Availability Statement:** All datafiles are available from Zenodo repository under the DOI 10.5281/zenodo.10043188, or through this URL: https://doi.org/10.5281/zenodo.10043188.

## Abstract

The Chesapeake Bay watershed is representative of governance challenges relating to agricultural nonpoint source pollution and, more generally, of sustainable resources governance in complex multi-actor settings. We assess information flows around Best Management Practices (BMPs) undertaken by dairy farmers in central Pennsylvania, a subregion of the watershed. We apply a mixed-method approach, combining Social Network Analysis, the analysis of BMP-messaging (i.e. information source, flow, and their influences), and qualitative content analysis of stakeholders' interviews. Key strategic actors were identified through network centrality measures such as degree of node, betweenness centrality, and clustering coefficient. The perceived influence/credibility (by farmers) of BMP-messages and their source, allowed for the identification of strategic entry points for BMP-messages diffusion. Finally, the inductive coding process of stakeholders' interviews revealed major hindrances and opportunities for BMPs adoption. We demonstrate how improved targeting of policy interventions for BMPs uptake may be achieved, by better distributing entry-points across stakeholders. Our results reveal governance gaps and opportunities, on which we draw to provide insights for better tailored policy interventions. We propose strategies to optimize the coverage of policy mixes and the dissemination of BMP-messages by building on network diversity and actors' complementarities, and by targeting intervention towards specific BMPs and actors. We suggest that (i) conservation incentives could target supply chain actors as conservation intermediaries; (ii) compliance-control of manure management planning could be conducted by accredited private certifiers; (iii) policy should focus on incentivizing inter-farmers interaction (e.g. farmers' mobility, training, knowledge-exchange, and engagement in multi-stakeholders collaboration) via financial or non-pecuniary compensation; (iv) collective incentives could help better coordinate conservation efforts at the landscape or (sub-)watershed scale; (v) all relevant stakeholders (including farmers) should be

**Funding:** This study was co-funded by The Pennsylvania State University – Institute for Energy and the Environment (https://iee.psu.edu/home), and by the LANDMARK (LAND Management: Assessment, Research, Knowledge Base) project (https://landmarkproject.eu). LANDMARK has received funding from the European Union's Horizon 2020 research and innovation programme under grant agreement No 635201. IRB approval was obtained through Penn State study #00011878 by author Caitlin Grady. The funders had no role in study design, data collection and analysis, decision to publish, or preparation of the manuscript.

**Competing interests:** The authors have declared that no competing interests exist.

concerted and included in the discussion, proposition, co-design and decision process of policy, in order to take their respective interests and responsibilities into account.

# I. Introduction

## 1.1. Background

Pennsylvania, with its strong historical and cultural connection to farming, benefits from a large agricultural sector worth $81.5 billion of direct economic output and generating 301,900 direct jobs [1]. The sector has been facing the challenge of remaining competitive in a global market while also reducing its ecological footprint for several years. Despite the State's efforts to reduce agricultural externalities, agriculture is still identified as a major cause of water pollution [2, 3].

The Chesapeake Bay, the largest estuary in the United States (US), has faced widespread water quality challenges, showcased by numerous classifications as "impaired" by the US Environmental Protection Agency (EPA) [3–5]. In 2017, it was estimated that 42% of the nitrogen, 55% of the phosphorus, and 60% of the sediments entering the bay were from agricultural nonpoint source pollution [6]. Consequently, the EPA has established a watershed-wide Total Maximum Daily Load (TMDL), a monitoring and management plan focusing on restoring clean water [4]. Despite positive results, a large part of the watershed remains on the EPA's list of polluted waters [3].

As societal pressure to reduce agriculture externalities has increased, concepts binding agriculture with environmental preservation and natural resources management have influenced discussions around agricultural policy reform in recent years. Notions such as multifunctionality of agriculture [7, 8], provision of ecosystem services [9, 10] and optimization of soil functions [11–13], attribute not only a primary production value but also ecological functioning values to agriculture. This relatively new approach to valuing agriculture is being increasingly institutionalized across the world [14–17]. In the US, agri-environmental policies combatting nonpoint source pollution are largely based on financial compensations of voluntary conservation practices, rather than on regulatory measures [14, 18, 19]. To date, these policy interventions have yielded insufficient results as in 2022, 45,000 km of waterways and 28,000 lake ha were still classified as "impaired" by the EPA [3]. Thus, the watershed's states are under pressure to comply with meeting the TMDL water quality requirements to reduce nonpoint source pollution from nitrogen, phosphorus, and sediments, by 36%, 30%, and 29% respectively by 2025 [6].

This research focuses on the information exchange around conservation practices on small and medium-scale dairy farms in central Pennsylvania. This sub-region of the Chesapeake Bay watershed (Fig 1) is representative of governance challenges relating to agricultural nonpoint source pollution and, more generally, of resources governance challenges in complex multi-actor networks settings.

Central Pennsylvania is an important livestock raising region, with half of the farms in Centre county being cattle farms, one third of which are dairy [20]. Several studies put forth dairy farmers potential contribution to meeting the local TMDL target from small scale dairy farmers in the region [21, 22]. In the following section, we present the theoretical framework describing the guiding concepts of our research. We then present the mixed methods approach used to collect and analyse both quantitative and qualitative data to understand information exchange around management practices for small and medium scale dairy farms in central

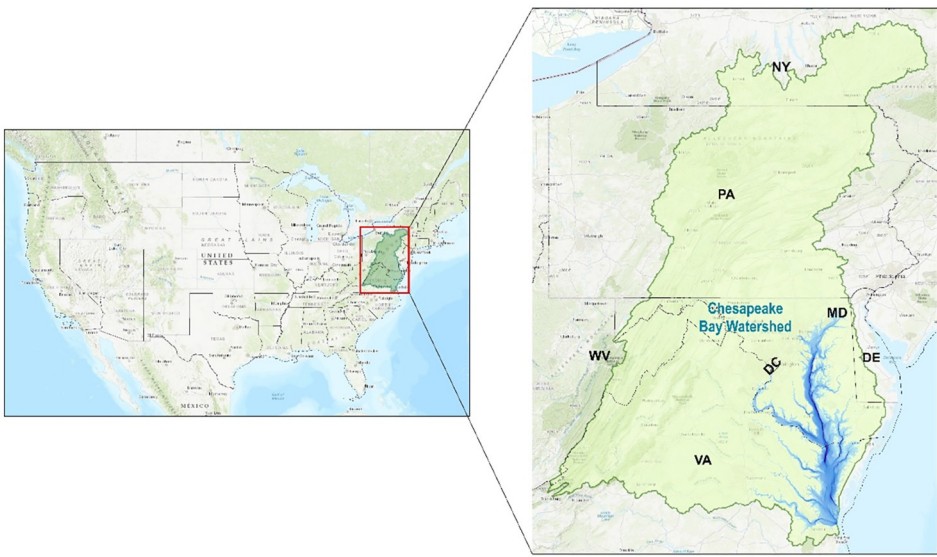

**Fig 1. The Chesapeake Bay watershed.** This map was created by the authors using publicly available shapefiles from the Homeland Infrastructure Foundation-Level Data (HIFLD) database.

Pennsylvania. In the results section, we disclose the resulting actors' networks and the distribution and strength of information flows, as well as insights on governance gaps and opportunities based on actors' perspectives. Finally, we discuss strategies to optimize the coverage of policy mixes, to reduce nonpoint source pollution from dairy farming in the study area.

## 1.2. Theoretical framework

Agricultural Best Management Practices (BMPs) are farm practices that generate public benefit by targeting nonpoint source pollution reduction [23, 24]. BMPs have been shown to effectively reduce nutrients runoff from farms, thereby contributing to restoring water quality [22, 25, 26].

Financial support, values and belief systems such as nonfinancial (stewardship) motivations or (dis)trust in science and institution, farm characteristics (farm size, vulnerable land), operators characteristics (trainings and education level), economic factors (level of income, capital, engagement in marketing arrangements), as well as access to quality information and connection to empowering networks, all influence BMPs adoption [23, 27–30]. Schall et al. (2018) conclude that "*the amount of information a respondent has about BMPs is less relevant than how this information comes to be interpreted and understood*". They highlight the necessity of considering the cultural and political frames that shape the farmers understanding, representations and opinions in designing educational programs [27], which also applies to the design of incentivization schemes and of governance mechanisms in general. This echoes Stuart et al. (2014) and Ulrich-Schad et al.'s (2017) adoption studies, that suggest that the information itself may be less decisive to a BMP adoption, than its origin (from where/whom the information was emitted) and the pathway through which it reaches the farmer. In fact, several network studies show that efficient (and effective) exchange of complex information depends on the extent to which the intervention (in our case, the diffusion of the information) is adapted to the topology (or configuration) of the network [31–34]. They emphasize that information flows need to be tailored to farmer types and farm systems specificities to enable innovation uptake and successful information exchange.

For example, targeting trusted brokers in a network, increasing the trust of farmers towards central information senders, or reducing the number of information-sources (by bundling information flows) to avoid information overload, are possible strategies to enhance the effectiveness and efficiency of an information-network [35–38]. This relates to the concept of "latitude of acceptance" in the Social judgement Theory, in which messages acceptance depends on the perception and evaluation of the message by the actor receiving it [39]. The receiver of the message will judge the message by comparing it to his/her current standpoint, thereby locating it in one of the three "zones of acceptance": the latitude of acceptance, of rejection, or of non-commitment [39–41]. The latitude in which the message falls, and thus its acceptance, is conditioned by a multi-layered judgement process relative to the current point of view of the receiver. Layers of acceptance include the acceptance of the underlying problem, the acceptance of the proposed solution, and the relationship and perception of the sender with/by the receiver (credibility and trust), among others [42, 43].

This highlights the importance of message credibility, and thus of their diffusion through strategic sources/senders in order to have greater chances to be accepted. Typically, farmers who are wary of the government, may automatically reject messages from government agencies [44–46], regardless of the content of the message. Thus, BMP-message acceptance by those farmers may meet less resistance if communicated via more credible senders, and thus may increase chances of BMPs adoption [46].

In the context of BMPs information in central Pennsylvania, it is crucial to understand how the actors network is structured, in order to determine how information diffusion can be adapted to the network. Tailoring agri-environmental governance mechanisms to the diversity of demands, motivations and attitudes that shape the farming profession calls for mixed development models (where various production and marketing systems are combined), underlining the need for pluralistic governance mechanisms [28, 47–49]. Stakeholders (other than farmers) throughout the food supply chain, as well as the organizations composing the social network around farmers, also have agency in addressing nutrient pollution issues [28, 47, 50, 51]. Understanding information exchange across these stakeholders, as well as their positions in the network, could unveil ways for more successful information exchange and for innovative governance strategies. Comprehending the network structure and how information is valued by farmers, can inform effective targeting of policy interventions, for example by identifying tailored policy entry points.

We refer to BMPs-information fluxes between actors as persuasive "messages", representing triggers or signals that can potentially lead to action undertaking or behavioural change [37, 42, 43, 52], in our case to adoption of BMPs. We conceptualize "messages" as interactions, or information fluxes tied to a specific topic (here BMPs), that actors exchange and that could elicit a response having implications for the said topic. The messages populating the governance context around an issue (here BMPs), inform and influence the decision-making process of actors (here land-users) at a local scale (e.g. at farm-scale) [37, 48]. Hence, here messages designate information flows with consideration for their influence strength (perceived credibility). In the context of our study, BMP-messages may influence the adoption (or not) of those BMPs by farmers. These messages can consist of actual information (news), knowledge, money (e.g. a subsidy, a grant), or any other interaction between actors, that relate to BMPs.

Social Network Analysis (SNA) offers methodological approaches for assessing these messages and their flow, to inform policy design and intervention. It is a tool commonly used to investigate social structures through the use of graphs theory, by determining key actors, their roles, their interactions and influences [37, 53–55], and has recently gained interest from the field of applied policy and agricultural development [37, 56–59].

### 1.3. Knowledge gap and research objective

In the case of Chesapeake Bay's nonpoint source pollution issue, the network and flow of BMPs-related information have not been studied. Increasingly, policy analysis highlights the need for policy mixes where instruments are optimally combined to reach policy targets [60–63]. Relations, complementarities, and effects of agri-environmental policy instruments have been widely assessed in literature [6, 62–64]. Less focus has been put on determining the actors that they should involve related to their existing position in the governance context. In fact, the targeting of entry-points based upon the network potential, the ultimate actors targeted by the intervention, and the pathways through which those can be reached have, to date, received limited investigation. Recent examples of such studies include the work of O'Sullivan et al. (2022) and Valujeva et al. (2022), which sought to identify key (strategic) actors in farmers networks. However, both studies focus on information flows relating to the concepts of soil functions, and do not address entry points and pathways for farm practices (such as BMPs) diffusion [37, 59]. This research contributes to filling this gap. Using SNA and the analysis of BMPs messaging to detect governance opportunities, we explore potential entry points for intervention to foster sustainable land management through improved BMPs communication. The method is tested on information exchange around BMPs adoption by dairy farmers in central Pennsylvania, with the following research question: How are farmers exchanging information about BMPs and can this exchange be captured and modelled using social network analysis to inform future policy interventions targeting?

Our hypotheses are that the information exchanged with farmers differs by actor and BMP type and that farmers value information differently based on actor.

Owing to the context specific nature of SNA, identifying universal schemes and generalizable policy guidelines is not appropriate, but rather, we want to demonstrate if/how different points of intervention can be optimally combined in policy mixes within the context of our study area to reach a wider range of farmers, and if strategies emerge to better integrate other (intermediate) stakeholders in policy intervention.

## II. Materials and methods

### 2.1. Mixed-methods and sampling procedure

The data for this research was collected following a mixed-methods approach combining surveys and interviews, which were conducted between February and April 2019. The survey provided quantitative and qualitative data relating to BMPs information exchange (messages), while the interviews provided complementary qualitative data about stakeholders' perceptions and contextual governance challenges and opportunities. The surveys and interviews were conducted with the same group of participants. These were identified by applying a snowball sampling method [65, 66], where the answers of the respondents oriented us towards the next actors to interview (Table 1). Some respondents were a part of the same organization; thus, the number of respondents exceeds the number of organizations.

The analysis process encompassed three distinct phases:

1. SNA, which analysed information flow and key actors in the network;

2. BMP-messaging analysis, which characterized the frequency, strength, and the distribution of BMP-related messages;

3. Thematic content analysis of qualitative data to assess governance challenges and opportunities.

**Table 1. Number of organizations interviewed (per organization type).**

| Type of Organization | | | Number of Organizations | Number of Respondents |
|---|---|---|---|---|
| Farmers* | Small-scale[1] | | 6 | 9 |
| | Medium-scale[2] | | 1 | 1 |
| | Large-scale[3] | | 2 | 2 |
| Local government (below state level) | | | 4 | 5 |
| Government (state or above state level) | | | 4 | 4 |
| Public institution of higher education | | | 1 | 3 |
| Non-profit organizations | | | 4 | 5 |
| For profit organizations | | | 3 | 3 |
| **Total** | | | **25** | **32** |

[1]Small scale: farm with <200 animals

[2]Medium-scale: farm with >400 and <1,000 animals

[3]Large-scale: Concentrated Animal Feeding Operation of >1,000 animals (CAFO)

*all farms are dairy operations, except of one small-scale farm (previous dairy, now crops).

Phases 1 (SNA) and 2 (BMP-messaging analysis) draw on the survey results, while phase 3 (Thematic content analysis) draws on the qualitative interviews outcomes. All data were collected in accordance with approval by the Penn State Institutional Review Board for study #00011878.

**2.1.1. Quantitative data.**   *Survey.* A preliminary review of scientific literature and informational discussions with experts guided the questionnaire formulation (S1 File), in particular the determination of a set of 16 BMPs (S1 Table). For each selected BMP, quantitative and qualitative data relating to message fluxes were collected, namely information about to whom the actor sent and received messages relating to the listed BMPs, as well as the type and the strength of the information (for messages type categorization refer to S2 File).

*Social network analysis.* The SNA was conducted on network graphs of BMP-message fluxes, using Gephi software [67, 68] for both visualization and calculation of network properties (Table 2). The graphs are composed of "nodes" (or vertices) representing the organizations (actors), and of "edges" representing their ties. These ties, or connection between nodes, correspond to the messages reported by the survey respondents, and are directed from the emitter of the message (source) towards the receptor of the message (target). Duplicate messages were merged into single edges.

Two network visualizations were constructed:

1. A socio-centric network graph, based on the full messaging dataset from all the survey respondents and representing the larger governance structure (including actors who may not interact directly with farmers);

2. A farmers-centric network graph, constructed with data from farmers-respondents exclusively, and composed of the farmers themselves and of the actors from whom they directly receive BMP-messages.

In order to protect the participant's identities, the actors and organization names were replaced by numerical pseudonyms in the network graphs. A complete list of the organizations that populate the nodes is presented in S2 Table.

*The socio-centric network.* All organizations that were interviewed, and all those identified by respondents as message senders, constituted a node of the socio-centric network (Table 3). The participating farmers, and the farmers to whom other respondents referred to, were

**Table 2. Description of network property measures and node's centrality measures in SNA [53, 69–74].** In the equations, *v* represents each vertex (node), *e* represents each edge. For Network diameter, $a(s, t)$ denotes the number of edges in the shortest path from a node *s* to a node *t*. For degree centrality, *i* is the transmitter node of *e*, *j* is the receiving node of *e*. For closeness centrality, $d(v, t)$ is the geodesic distance between any node *v* and *t* (i.e. the sum of the edges on the shortest path). For betweenness centrality, $g_{st}$ represents the number of geodesics (or shortest paths) connecting *s* to any other node *t* in the network, and $g_{st}(v)$ denotes the number of shortest paths from *s* to *t* that some node *v* lies on. For clustering coefficient, $n_v$ is the number of alters (neighbours) of *v*, and $m_v$ is the number of alter-to-alter edges in the neighbourhood of *v*.

| Measure | Equation | Definition | Explanation |
|---|---|---|---|
| Network diameter | $D = \max_{st}\{a(s, t)\}$ | The shortest path length in the network (the shortest distance between the most distant nodes in the network). | Indicates how long it would take (or how many intermediary nodes it would take) for information to circulate between the two most distant nodes in the network. |
| Degree (in-degree, out-degree) | In-degree $IDi = \Sigma\|e_{ij}\|$ k = 1 Out-degree: $ODi = \Sigma\|e_{ij}\|$ k = 1 | The number of relations (edges) of the nodes. In directed graphs, a distinction is made between in-degree (number of incoming edges) and out-degree (number of outgoing edges). | Effective measure to assess the importance of an actor in a social network, but doesn't take into consideration the global structure of the network. In-degree: frequency of message receiving. Out-degree: frequency of message sending. |
| Closeness centrality | $C_C(v) = \frac{1}{\Sigma d(v,t)}$ | The average length of all shortest paths from one node to all other nodes in the network. The farness of a node is defined as the sum of its distances from all other nodes, and its closeness is defined as the reciprocal of the farness. | Broadcasters, measure of reachability: closeness to the entire network, i.e. how easily a node can reach all the other nodes in the network (ability to reach the entire network quickly, to broadcast information). |
| Betweenness centrality | $C_B(v) = \Sigma\, g_{st}(v)/g_{st}$ | The extent to which a node lies between other nodes in the network. The fraction of shortest paths that go through a node divided by the total number of shortest paths between all nodes. | Brokers or gatekeepers: interfaces between tightly-knit groups, nodes with high a high betweenness centrality are vital elements in the connection between different regions of the network, and can control the flow of information between communities. |
| Clustering coefficient | $C_{Cl_v} = m_v / \frac{n_v(n_v-1)}{2}$ | The fraction of the possible relation triangles that are actually completed. A node's (local) clustering coefficient is the fraction of its possible relational triangles that are actually completed. The (global) network clustering coefficient (computed for the whole network), is the measure of all completed relational triangles over all the possible relational triangles in the network. | Node's clustering coefficient: The extent to which a node's direct neighbours or alters (i.e. nodes to which it is connected) are also likely to be neighbours (connected). Indicates the level of cohesion between the neighbours of a node. |

merged into a broader node labelled "Farmers". The network boundaries were determined when saturation was reached, based on two observations: (i) when pursuing the snowball sampling added duplicate edges, and/or only added nodes that would have no 1st or 2nd degree connection to farmers; (ii) when all the key actors (listed prior to the sampling process, based on literature and informational discussions with experts and key-informants) appeared in the network.

**Table 3. Actors categorization and related colour code.**

| | |
|---|---|
| **Farmers** | farmers, agricultural land managers and/or landowners |
| **Private for profit** | private agricultural enterprises including agricultural inputs suppliers, private consultants, veterinarians, animal nutritionists, trade associations |
| **Non-profit / NGO** | private non-profit and Non-Governmental Organizations including environmental and social advocacy groups and farmers associations |
| **Local government** | local government agencies with below state level outreach (e.g. central Pennsylvania, county, township, or borough level) |
| **Government (≥state)** | state or above-state government agencies with outreach at state, multi-state or federal level |
| **Universities** | public institutions of higher education |
| **Other** | multi-actor partnership (e.g. public-private partnership, consortiums), the media, civil society, primary or secondary education |

*The farmer-centric network*. The farmer-centric network includes the farmers-respondents themselves (as individual nodes), as well as the actors they reported receiving information from. The other actors were attributed the same numerical pseudonyms and colour code as in the socio-centric network. An additional node '*Other farmers*', representing the rest of the farmer community, was also added to the network.

The edges, or ties, of the farmers-centric network represent the messages received directly by farmers, hence only those reported by the farmers-respondents were included.

In order to compare BMP-specific networks, the farmer-centric graph was constructed for four different situations: (i) considering the messages for all BMPs, (ii) considering the messages relating to (forested and grass) riparian buffers exclusively, (iii) considering the messages relating to no-till and cover-cropping exclusively, and (iv), considering the messages relating to manure management exclusively (having a manure management plan, following a manure management plan, and manure storage). These categories were based on grouping of the most important BMPs into BMP-types: riparian buffers, soil conservation cropping practices, and manure management.

*BMP-messaging analysis*. The BMP-messaging analysis first focused on the frequency and distribution of all messages reported by the survey respondents (after merging of duplicates), considering the BMP they were tied to, and the type of information they related to.

A second analysis was performed on the messages from the farmers-centric network (messages received and reported by farmers directly). Therefore, the comparative frequency and distribution of the messages weights were analysed considering the BMP they related to, and their source (actor who emitted the messages). Duplicate messages were merged, and their weights averaged.

**2.1.2. Qualitative data.** Additional qualitative data was collected via semi-structured interviews, guided by open-ended questions relating to governance challenges and opportunities of nonpoint source pollution management (S3 File).

The interviews notes and audio-recordings were transcribed into text format, on which inductive content analysis was conducted through a systematic coding process [75–78]. The interviews coding process was split into three stages, using QDA Miner Lite software [79]: decontextualization, recontextualization and categorization [75, 77, 80].

The textual data from interviews was first disassembled (decontextualization) by systematically coding repeated observations. Sections of text were coded if their content reflected ideas or concepts mentioned in previous interviews (repetition), if they were explicitly indicated as important, or/and if they echoed with information from literature or discussions with experts. During the recontextualization process, the interviews were re-read and recoded in order to refine the codes (adding, merging or deleting of initial codes). At the same time, coded text sections were tagged:

- with "H" (= hindrance) if the information was explicitly cited as a governance issue, a challenge or a hindrance, or appeared to be a factor that prevented BMPs adoption or nonpoint source pollution reduction;

- with "O" (= opportunity) if the information was explicitly cited as an opportunity or a possible solution, or appeared to be a factor that allowed for/favoured BMPs adoption or NPS pollution reduction.

The data were then reassembled, by grouping the codes into concepts/themes. The frequency of themes and codes, as well as the frequency and distribution of tagged codes were then analysed, in order to determine the relative importance of the governance challenge and opportunities from the stakeholder's perspective.

**Table 4. Graph properties of the socio-centric network calculated on Gephi.**

| Nr of edges | 3919 |
|---|---|
| Nr of nodes | 57 |
| *Farmers* | *1* |
| *Non-profit* | *17* |
| *For profit* | *6* |
| *Local government agencies* | *8* |
| *Government ($\geq$state)* | *15* |
| *Universities* | *1* |
| *Other* | *9* |
| Network diameter | 5 |
| Average path length | 2.3 |
| Average degree | 68.8 |
| Average clustering coefficient | 0.24 |

# III. Results

## 3.1. The socio-centric network

**3.1.1. Graph properties.** The socio-centric network is composed of 3919 edges (messages) and 57 nodes (actors or organizations), over half of which are non-profits or state (or above state) government agencies. Of the 9 nodes falling under the category "Other", 6 correspond to multi-actor partnerships (Table 4).

The graph diameter (5) indicates a rather small but highly connected network [70]: in order for BMPs messages to be exchanged between the two most distant actors of the network, the information would have to transit by four intermediary actors. The average degree of nodes (68.8), indicates that a typical actor of the network exchanges (receives or sends out) as many as 69 BMP-messages in average per year [58]. The fact that the network diameter is nearly twice the value of the average path length, suggests a high communication efficiency across the network. In addition, the average clustering coefficient (or average neighbourhood completeness of nodes), reveals that nearly 1/4th of the possible relation triangles between three nodes are completed, indicating that a fairly high level of cohesion is present in the network (Table 4) [72].

**3.1.2. Node properties.** A few actors repeatedly scored the highest centrality measures. The most active BMP-message senders are also the most important message receivers in the network, but in different orders. For example, farmers (node 1) only rank at the 5th position as message senders (Fig 2A.i and 2A.ii), while they are the most important BMP-message receivers (Fig 2B.i and 2B.ii). On the other hand, the top-3 message senders are the two most prominent state (or above-state) government agencies (nodes 39 and 14) and an NGO (node 44), while a private for profit organization is the third most important message receiver (node 48), after farmers (node 1) and a government agency (node 14). This indicate that most messages converge at farm level, and that farmers are (one of) the main information receivers.

The betweenness centrality measures (Fig 2C.i and 2C.ii) show that the active message receivers and senders often also act as the shortest path between pair of nodes that are not connected, thereby acting as bridges (brokers) between actors [53, 73]. In fact, the two prominent government agencies (nodes 14 and 39) are also in the top-5 message receivers and senders, followed by the aforementioned NGO (node 44) and farmers (node 1). Public institutions of higher education (node 7: Universities), show a nearly as high betweenness centrality measure than the later, and are the fifth major broker of the network. These actors are vital elements of

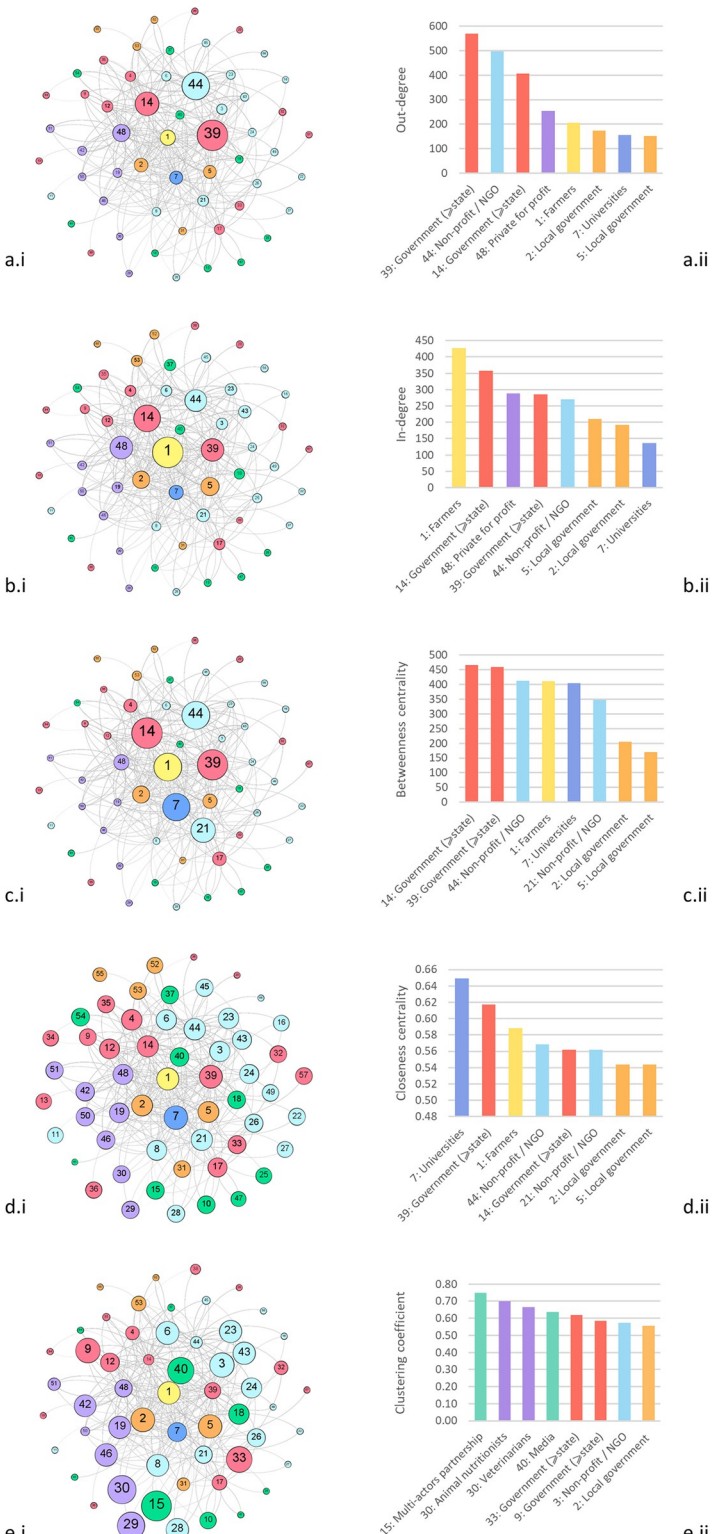

**Fig 2. Socio-centric networks.** a = out-degree, b = in-degree, c = betweenness centrality, d = closeness centrality, e = clustering coefficient; i = SNA graphs, ii = bar graphs showing the top eight centrality measure scores (Y-axis) of nodes (X-axis); the color code reflects the actors type classification.

the network, as they act as intermediaries to connect areas of the network that would otherwise be disconnected. Universities (node 7) also record the highest closeness centrality measures (Fig 2D.i and 2D.ii), suggesting that they are the most connected actors of the network, followed by one of the prominent government agency (node 39) and farmers (node 1). The closeness centrality measure displays less heterogeneity across nodes of the network as the other centrality measures.

New actors appear in the top-4 clustering coefficient measures results (Fig 2E.i and 2E.ii), namely: a multi-actor partnership (node 15), two for profits relating to animal services (nodes 30 and 29), and the media (node 40), which suggests the existence of sub-networks or "small worlds" within the larger network [70]. High clustering coefficients are an indicator of tightly knitted neighbourhoods (cliques). These nodes with a high clustering coefficient may be worth targeting to relay information to their neighbours while keeping the number of links to connect the network to a minimum [37].

## 3.2. The farmer-centric network

### 3.2.1. Graph and node properties.
Farmer-respondents reported having received messages from 21 actors over the 57 actors of the socio-centric networks (37%), resulting in a farmers-centric network of 346 edges (messages) and 30 nodes (including the 9 farmers who were interviewed) (Fig 3A), with an average degree of 11.5. Degree of nodes refers to in-degree (received messages) for farmers nodes, and to out-degree (emitted messages) for the other nodes. Farmers in-degree vary considerably, ranging from 16 (node 58) to 102 (node 64), illustrating the high variability in farmers exposure to BMPs information.

When considering all messages, without distinction between BMPs, the most active message senders are two local government agencies (nodes 2 and 5, out-degrees = 65 and 60, respectively), followed by private consultants (node 42, out-degree = 34), and the media (node 40, out-degree = 29) (Fig 3A).

However, this order changes when considering messages relating to specific BMPs, showing that different (types of) actors communicate on different types of practices.

For example, the most active message senders for riparian buffers remain the local government agencies, followed by an NGO (node 3) and a state government agency (node 12) (Fig 3B). On the other hand, no-till and cover-crops messages appear to be more strongly tied to other farmers (node 1), the media (node 40), a local government agency (node 2) and agricultural industries (node 19), in descending order (Fig 3C).

When looking at manure-management messages (manure management planning and manure storage), the same local government agencies stand out (nodes 2 and 5), followed by private consultants (node 42) and a state government agency (node 12). The larger farms (nodes 64, 65 and 66) receive more messages relating to manure management than the small-scale farms (nodes 58 to 63) (Fig 3D).

Our results show that the farmers reached by for-profit type of actors (light blue nodes) are different than the farmers reached by non-profit type of actors (violet nodes) (Fig 3A). Furthermore, only peer farmers (node 1) reach all types of farmers in the network (Fig 3A), but these information fluxes relate to specific types of BMPs (rather agronomic and cropping practices such as cover-cropping and no-till), and exclude other BMPs such as riparian buffers (Fig 3A and 3B), while they are very limited for manure management related practices (Fig 3D).

The most prominent nodes of the socio centric network (nodes 39 and 14, 44 and 48), and most state (or above state) government agencies, do not appear in the farmer centric network, implying that they are more active in the macro governance structure than in direct interactions with farmers. In contrast, the local government agencies that are most active in the socio-

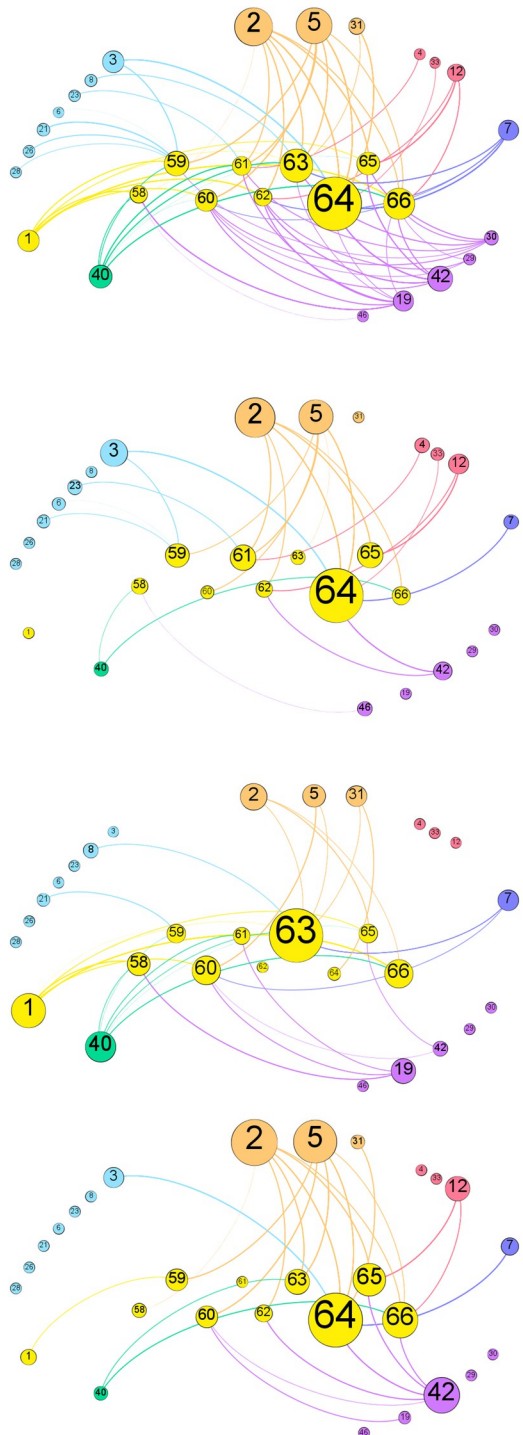

**Fig 3. The farmers centric network.** Nodes' color code relate to the actors' type classification; a = all BMP-messages; b = forested and grass riparian buffers messages; c = no-till and cover-crops messages; d = manure management plan messages. Node sizes are relative to the node's degree, and edge colors correspond to the source (i.e. the actor emitting the message). Nodes 58, 59, 60, 61, 62, 63 = small-scale farms; node 64 = medium-scale farm node 65 and 66 = large-scale farms (CAFOs).

centric network (nodes 2 and 5), as well as universities (node 7) and the media (node 40), are also important in the farmers-centric network. This indicates that these are active actors in both, the macro governance structure as well as on the ground in closer interaction with farmers. Hence, local government agents, universities and the media play important bi-directional top-down and bottom-up brokerage roles. Private non-profit organizations with boards constituted of a majority of farmers ("farmer led organizations") are mainly active on the ground (farmer centric network), but receive a lot of messages from the macro level too, therein acting as top-down and horizontal information channels.

### 3.3. BMP-messaging analysis

**3.3.1. Message characterization and distribution.** In total 4344 BMP-messages were reported by the survey respondents, after merging of duplicates. The BMPs for which the most messages were reported (>350) are cover crops (395), forested riparian buffers (385), no-till or conservation tillage (372), in descending order (Fig 4).

The BMPs for which the least messages were reported (<210) are reduced stocking density (207), precision fertilization (184) and hedgerows plantation (127). Over half of the total information flow related to knowledge exchange or knowledge dissemination; nearly one quarter (22%) related to regulation or standards; 11% to funding; 10% to personalized technical assistance to implement BMPs (assistance request or provision), and only 5% to networking and actors linkage.

**3.3.2. Message strength.** In the farmers centric network, BMP messages are most numerous for 'Having a manure management plan' (40 messages reported), 'Forest riparian buffers' (35) and 'Manure storage' (34); and are the least frequent for 'Hedgerows plantation' (0), 'Precision fertilization' (7), 'Wetland and permanent grassland preservation' (10), 'Following a nutrients management plan' (13) and 'Following a manure management plan' (13) (Fig 5). Hence, the relative importance of BMPs (in terms of message frequency) are different in the farmers-centric network and in the socio-centric network, suggesting that the focus on specific BMPs differs across governance scales. In fact, only forest riparian buffers messages are among

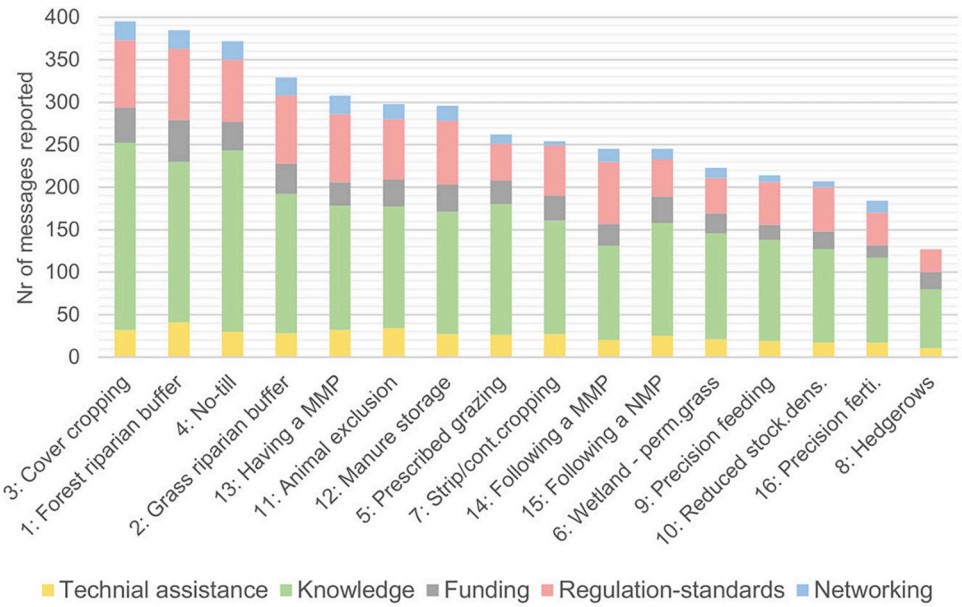

**Fig 4. Messages distribution per BMP (X-axis) and per message kind, after merging of duplicates.**

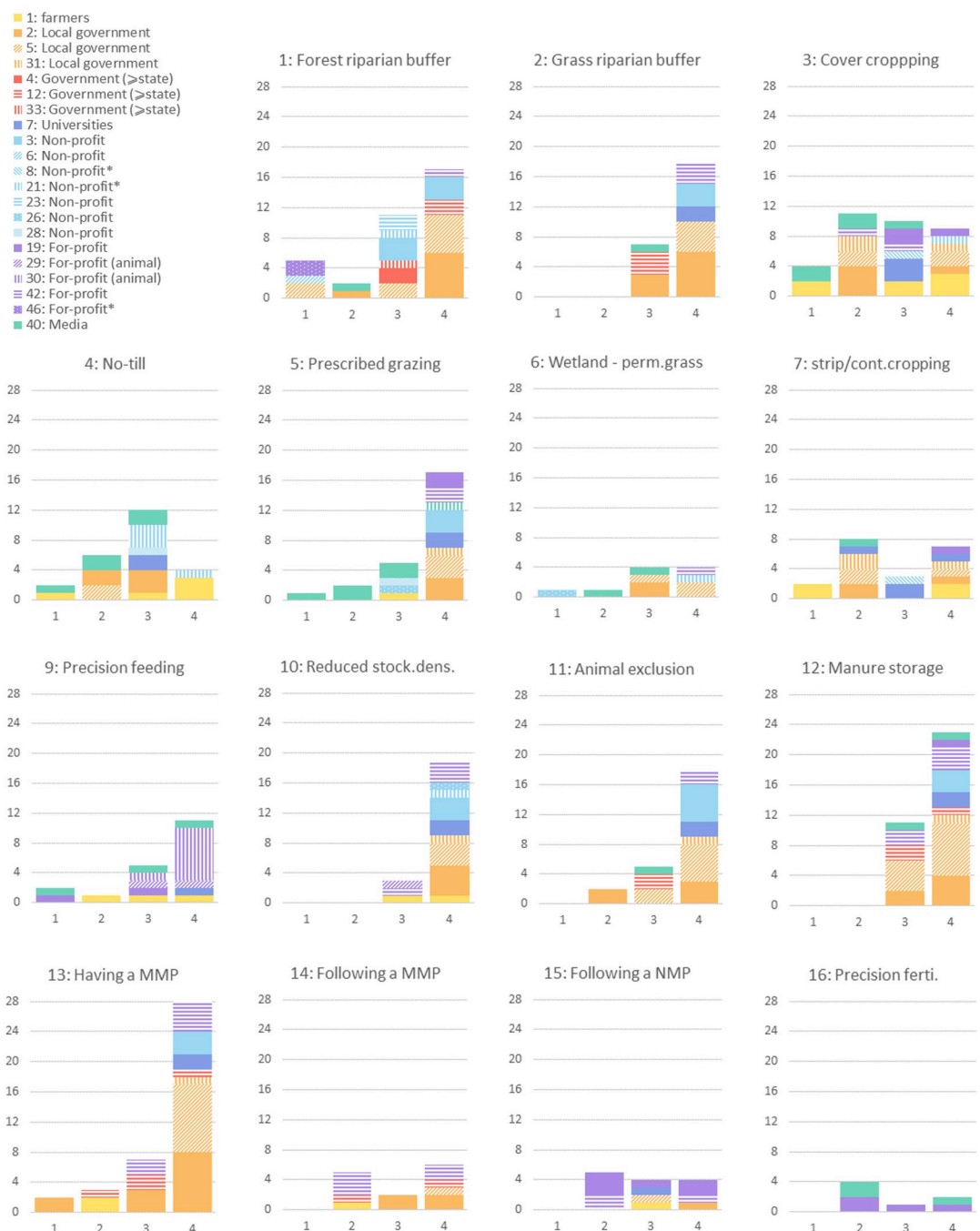

**Fig 5. Messages distribution per BMP (includes only the messages received directly by farmers).** X-axis: message strength based on influence-rank assigned by farmers (1 = weak, 4 = strong); Y-axis: messages count; legend: message source (emitter of the information); 'For-profit (animal)': animal specialist, '*': farmers-led organizations. Note: no messages were recorded for BMP nr 8 'Hedgerows plantation'.

the most prominent BMPs in both networks, while hedgerows plantation and precision fertilization messages are the least important in both networks.

The BMPs recording the highest frequency of strong influence (rank 4) are, in descending order: 'Having a manure management plan', 'Manure storage', 'Grass riparian buffers' and 'Reduced stocking density', followed by 'Animal exclusion' (Fig 5). Farmers assigned high

influence strengths to these BMPs, while they do not seem to communicate much about them among peers. In contrast, farmers seem to share more information about agronomic and cropping practices such as no-till, cover cropping and strip-cropping; which are also strongly influenced by farmers-led NGOs, i.e. private non-profit organizations, the boards of which are composed primarily of farmers (Fig 5).

The sources of the messages (actors who emitted the information) recording the highest frequency of strong influence (rank 4) are, in descending order: local government agencies (nodes 5 and 2), a private non-profit organization (node 3), and private consultants (node 42) (Fig 5).

While most BMPs are tied to many different actors, our results show that the influence of some actors is limited to certain BMPs. State and above state government agencies for instance, are only present when it comes to riparian buffers, manure management (planning and enforcement) and animal exclusion. The mandatory aspect of manure management planning may explain this strong link with state or federal authority, while the importance of riparian buffers here illustrates the keen interest in this practice at a higher governance level. Local governments are closer to farmers on the ground, and are influential agents for all BMPs, aside from precision fertilization and precision feeding. In contrast, these BMPs seem to be more tied to the profit sector, perhaps because of their reliance on inputs provision (fertilizers and animal feed). Furthermore, specific profit agents are influential on specific BMPs. For instance, not only are agricultural industries strong influencers on cover-cropping, but also on nutrients management planning, precision fertilization and prescribed grazing. Additionally, private consultants are influential on grassland management choices (grass riparian buffers, grazing and permanent grassland management), as well as on manure management (reduced stocking density, manure storage, having and following a manure management plan). Lastly, animal specialists such as veterinarians and nutritionists, as one might predict, are influential agents for discussing livestock density reduction precision feeding practices.

Although public universities (node 7) only reach a limited number of farmers (Fig 4), they are influential on a wide range of BMPs, and their messages were systematically assigned high influences by farmers (rank 3 or 4). Hence, not only do they play an important brokerage role in the overall governance structure, but are also influential actors on the ground.

Messages originating from the media are better ranked when relating to technology and technical equipment such as precision fertilization, precision feeding, manure storage. In contrast the media record rather low influence scores when relating to agronomic and cropping practices (no till, cover-cropping, strip cropping, as well as for grassland management (grazing) (Fig 5).

## 3.4. Thematic content analysis

The systematic coding of the interviews resulted in a set of 34 codes, that were grouped into 7 themes (Fig 6). A detailed list of the themes, codes, and their description is presented in S3 Table. The highest frequency codes for both hindrance and opportunity are showcased in Table 5.

The first major theme, "Perception of policies and actors", relates to stakeholders' perceptions of the roles/responsibilities of other actors (government, for-profit, non-profit, peer-farmers) or their relation to them. While government was frequently mentioned as (financial and technical) facilitators of BMPs implementation, farmers distrust towards the government was also stressed.

For instance, a farmer explained that connecting with local government agencies encouraged him to deploy BMPs on his farm, while (some) neighbouring farmers were much more

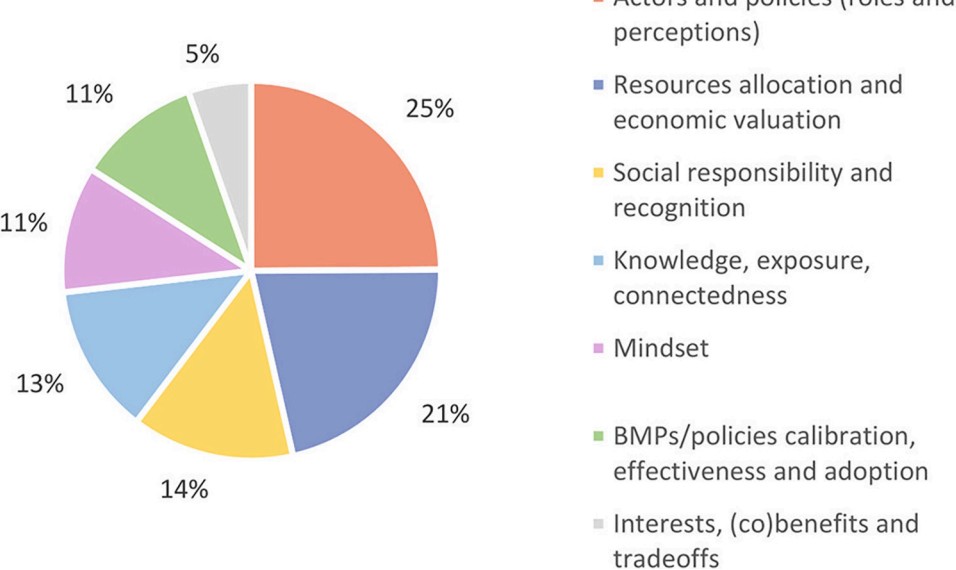

**Fig 6. Relative importance of the emerging themes from the interviews content analysis.** Each theme refers to a cluster of codes, the frequency of which were computed to establish the relative importance of the theme (%).

wary: *"(. . .) what got the ball rolling (. . .), it started with the grant. Although, (we were) doing some of this stuff ahead of time. But when (the local government agency) came in, and we formed a relationship with them, it became pretty easy after that to take their recommendations and transition with their suggestions. (. . .) And we saw the one programme work. So we tried another. (. . .) So, it just kind of snowballed from there. (. . .) some of our neighbours (. . .) don't want the government around. They don't trust them. They think if they're coming it's for something bad. And they're probably not the only ones that have that attitude."* Another respondent also clearly stated his aversion: *"I have no interest in having any government money for inputs. I know what I'm doing."* In his opinion, the government is deliberately letting small-scale farms disappear: *"the government is trying to force out all small farmers. Get rid of all small dairy farmers. (. . .) Because they want to just control a few dairies. Quite big dairies. (. . .) It's easier to monitor 2 dairies that have 10,000 cows than a hundred dairies that have 50 cows."*

This theme also relates to the participants' (positive or negative) critical perception of (regulatory or voluntary) policy measures. While some farmers felt constrained by regulation, others insisted on the need for more enforcement: *"Better enforcement of compliance with (agricultural) requirements is essential, and will drive improvement. Right now, in Pennsylvania, compliance is cursory at best, no real zeal behind it. Without the threat of some consequences for inaction, no change is the most attractive option"*. This opinion was shared by several non-farmer interviewees, while one mentioned conflict of interests as the main obstacle impeding enforcement. Another participant suggested that this could be addressed by establishing a dedicated revenue-source for BMPs establishment and enforcement: *"(. . .) if the Commonwealth establishes a revenue source, or commits so much money each year to create a revenue source to help pay for these practices, then I think you would see more of a political will to maybe start enforcing some of these laws. (. . .) But without a revenue source to do that, no enforcement takes place"*. Representative of the contrasting opinions among stakeholders, the code "regulation and enforcement" was cited as both an opportunity and a hindrance with similar frequencies. On the other hand, "voluntary measures" were considered as an opportunity as frequently as

**Table 5. Maximum frequencies of observations from interviews content analysis (Fq: frequency, Int: interview, edu.: education).**

| | | Codes occurrence | | |
|---|---|---|---|---|
| | Code | Fq. count | Int. concerned | % Int. |
| | funding | **109** | **21** | **81%** |
| | government | **107** | **20** | **77%** |
| | edu./knowledge and outreach/exposure | **92** | **22** | **85%** |
| | regulations and enforcement | 52 | 14 | 54% |
| | innovation, adaptability/flexibility | 48 | 18 | 69% |
| | viability | 47 | 15 | 58% |
| | diversity and (mis)calibration | 46 | **19** | **73%** |
| | responsibility distribution | 36 | 18 | 69% |
| | | Tags occurrence | | |
| | | Code | Fq. count | |
| **Hindrance** | | government | **31** | |
| | | funding | **28** | |
| | | cost-share | **24** | |
| | | regulations and enforcement | 21 | |
| | | viability | 19 | |
| | | priorities | 17 | |
| | | human and time resources | 17 | |
| | | diversity and (mis)calibration | 16 | |
| | | selectivity | 14 | |
| **Opportunity** | | funding | **69** | |
| | | government | **66** | |
| | | edu./knowledge and outreach/exposure | **62** | |
| | | innovation, adaptability/flexibility | 34 | |
| | | collaboration | 33 | |
| | | regulations and enforcement | 25 | |
| | | voluntary measures | 25 | |
| | | tradeoffs or co-benefits | 24 | |
| | | human and time resources | 24 | |

regulatory measures (frequency = 25), while they were rarely mentioned as a hindrance (frequency = 9) (Table 5).

The second major theme, "Resources allocation and economic valuation", relates to the availability of financial, human and time resources, to the economic valuation of environmental/social services provided by farmers, and to the viability of farms and their (in)capacity to self-finance BMPs. Farmers-interviewees emphasized that financial assistance was crucial in their decision/ability to implement BMPs on their farms, while government agencies highlighted that funds were, in general, not sufficient to cover farmers demands and to provide appropriate extension services. The poor investment capacity of farmers and their struggle to keep the farm viable was repeatedly mentioned as a main challenge, implying that farmers would not prioritize BMPs as long as it does not increase profitability in the short/medium term and help them keep their farm viable: "*right now, dairy farms are in an economic crisis. So, some can't even afford to feed their own families. (. . .) even when the milk price goes back up, producers will spend years digging out of this whole. So, they are not going to use that extra money to implement conservation practices*".

Interviewees recognized that farmers' role was undervalued compared to the results/outputs generated. Better economic valuation of environmental and social services provided by farmers was cited as a possible solution to address this challenge: *"if society was ready to pay the price associated with producing the food, given the costs to maintain the environment, the way that the other 90% of the population that's not involved in agriculture want to see the environment maintained, then they should probably pay 20, 30% more for their food"*; *"Chesapeake Bay (. . .) generates, what, like $6.8 billion in revenues as a result of clean water. (. . .) all these benefits, job creation, recreation, (. . .) value will increase, the cleaner the bay gets"*. One farmer expressed that we should *"measure success a little differently"*, for example by rewarding soil restoration.

The third theme, "Social responsibility and recognition", relates to power and responsibilities distribution, to social recognition, and to farmers (political) representation and inclusion in discussions and decisions. Sense of (un)fairness and unequitable distribution of pressure stood out as a sensitive subject in many interviews. For instance, one farmer said *"there's always things we can be doing to make improvements, and do a better job of what we're doing. But I don't think we're the sole reason there's a problem. (. . .) the general public doesn't recognize that. (. . .) if you're going to put us under the thumb, everybody else should be held to the same standard, and I feel like this is something we've been working on for a solid 20 years, where townships, a lot of these other communities are not held to the same standard we are"*. Relating to farmers' inclusion and consultation, this same farmer stated: *"it's important we need regulations, or may. . . (but) farmers need to be at the table. (. . .) one of the biggest frustrations is when you get a bunch of government and university people together, that all don't do this every day. And on paper, it looks like a really great logical solution, but the application in reality is very different. (. . .) Farmers will be more open to it, if it's done in a good way, and presented in a good way"*. Likewise, another farmer put forth the need to consult farmers and make use of their knowledge: *"their (= the farmers') voices aren't heard. (. . .) the people writing the regulations (. . .) haven't lived there and haven't experienced it, (. . .) they may be book smart but they are not necessarily*

street smart. (. . .) they write the laws (. . .) using general math, and don't necessarily use common sense."

Finally, the theme "Knowledge, exposure, connectedness" refers to education and knowledge exchange, to connectedness with other networks/actors, to collaboration and coordination between actors, and to farmers geographic (im)mobility (isolation or propensity to travel). The code "education/knowledge and outreach/exposure" recorded the third highest opportunity-tag frequency, and "collaboration" (i.e. multi-actor and multi-scale partnerships, coordination between actors) the fifth highest (Table 5). For instance, one of the participants highlighted the need for stronger inter-municipal and watershed-scale collaboration: *"more cooperation and collaboration between municipalities (is needed), because what oftentimes you see is an upstream problem causing (. . .) downstream water quality issues. (. . .) one thing that could possibly help is having multi-municipal, or having some kind of watershed impact be a part of funding requirements. (. . .) And getting input and support from other municipalities to see how this project is going to benefit others."*

Interestingly, "government" and "funding" are the most redundant codes (overall, but also in terms of both "opportunity-tags" and "hindrance-tags" occurrence), but were reported over twice more frequently as opportunities than as hindrances.

## IV. Discussion

We have shown that different actors communicate to farmers on different types of BMPs, that farmers exposure and networks vary considerably and that farmers may value information

differently based on its source. We also showed that intermediary actors play important brokerage roles according to their position in the macro-governance network. Drawing on the governance gaps and opportunities that we identified, in this discussion we articulate potential strategies that may provide opportunities for governance and management of non-point source pollution as well as several limitations of this study.

## 4.1. Targeting interventions based on network topology

The coverage of policy mixes may be improved by targeting a specific variety of messages sources, representative of the high variability in farmers' networks and exposure [46, 59]. Our SNA results suggest that government agencies are connected with all types of farmers (Fig 3), yet the results of the qualitative data analysis reveal that farmers' avoidance of government-related actors is common. We may have failed to capture the "mistrustful" farmers networks, as they may not be represented in our respondents' sample. Hence, although BMPs dissemination through government and universities can be effective, it may not be enough. In fact, studies show that farmers participation in research and government-led programs is limited to certain types of farmers, and that certain farmers are unlikely to ever participate to such programs [46, 81, 82].

Our SNA also found that some farmers are very connected to the for profit sector, which is increasingly viewed as a potential entry-point for BMP intervention [83, 84]. In fact, BMPs valuation through markets could present interesting opportunities given that government programs have several limitations including a lack of funding, mandate, or farmers reluctance to participate. This would also allow to target intervention and incentivization towards other actors of the supply chain and alleviate the concentration of pressure and information on farmers. This aligns with a recent European-level study [37], where targeting "higher" entry points for messages diffusion was identified as a strategy to reduce message overload on farmers. Furthermore, targeting intervention towards other (higher) actors in the network may improve messaging coherence and alignment through "bundling" of messages, i.e. grouping messages to reduce the number of information-sources [37], thereby reducing the "noise" perceived by the end-target actors (here farmers).

Contextualised to BMPs adoption in Pennsylvania, adaptations such as supply-chain perspectives (through sustainable and ethical sourcing for example) are promising avenues that have raised the interest of shareholders and investors and thus, are being increasingly adopted by agri-food businesses [85, 86]. Such initiatives are emerging in the region, such as the Turkey Hill Clean Water Partnership in Lancaster County, a collaborative effort between a dairy processor, milk producers cooperatives, and non-profit organizations, to incentivize conservation planning on farmland [87]. Scaling up such initiatives could have a massive impact, given the importance of Pennsylvania's food processing and manufacturing sub-sector, accounting for 60% of the state's agricultural economic output [88].

Several studies have shown that agricultural cooperatives can play an active role in encouraging sustainable farm practices [89–92], while government support programs can support these cooperatives through capacity building [93]. However, despite the density of BMP-messages flow in the network, none of the survey respondents has reported receiving or emitting BMP-messages to/from processors or dairy cooperatives, suggesting that these are, at this time, poorly involved in conservation efforts in central Pennsylvania. This could be addressed through incentivization schemes encouraging cooperatives and processors to adopt sustainability standards and supply-chain governance approaches. Certain private non-profit organizations, or state level government agencies could facilitate this process [94, 95].

Furthermore, private farm consultants may constitute possible entry points for manure management policies, revealing a potential role for private companies in compliance control.

For example, in the same way that private nutrients and manure management planners are certified by the state [51], compliance control of the actual application of these plans by farmers could be conducted by accredited private certifiers. This may be a more popular option among farmers, than compliance-control being conducted by local government agencies. In fact, farmers increasingly rely on private consulting firms [96, 97]. Interviewees reported that this may be due to distrust towards government and to reduced public resources (human and financial) for enforcement and extension [98–100]. Private farm advisors can also be conservation intermediaries [98], by providing technical assistance to farmers for BMPs implementation. This was considered a 'risky opportunity' by some interviewees, because of possible competition between public conservation interests, and the advisors' private economic interest, as also acknowledged in previous research [51, 98, 101–103]. Nonetheless, the role of the for profit sector in fostering BMPs adoption, and the extent of its outreach, should not be underestimated and should be considered when designing policy mixes. As participation in BMPs by certain farmers may be undermined by unalignment of their (relational) values with government-led programs [46, 104] public-private partnerships may offer opportunities for more effective policy and extension outreach [46, 83]. Our results show that most of the for profit organizations are in close interaction with farmers, but are less connected to the overall governance structure. They also reveal that some actors, such as farmers organizations, trade associations, local governments and universities can be bridges between the government and some of these "small worlds".

The SNA results suggest that farmers are the actors who are most likely to reach a wide range of other peer-farmers, which is congruent with the findings of Valujeva et al. (2022) and O'Sullivan et al. (2022) who identified peer-to-peer farmers interactions as a major communication channel in European countries. Hence farmers exposure should be strongly supported to foster inter-farmers interaction. Therefore, programmes focusing not simply on BMP adoption, but on farmers' coordination, mobility, training, and knowledge-exchange, could be developed. In fact, numerous studies showcase how programmes' impact and outreach were multiplied by inter-farmers interaction [31, 32, 105–108], and how collective action contributes to enhancing farmers participation and coordination [109–113]. Different approaches have been identified by Valente (2012) to frame interventions based on network topology. The most common one being the targeting of individuals such as opinion leaders (local champions), bridging individuals (who may be more amenable to change and diffuse change than leaders who have interest in maintaining the status quo), and actors at the periphery of the network (to avoid their exclusion, but also because they may be sources of innovation as they are likely to be connected to other communities or networks). Another approach is through segmentation (identifying groups of people to change at the same time), which may be useful to reach communities with established norms and processes that will only change if the whole group changes. Finally, induction interventions, such as word-of-mouth situations or media marketing campaigns, stimulate interpersonal communication persuading others to adopt a new behaviour. These interventions do not necessarily use network data, but depend on the network for their effects. Thus, induction is an effective strategy to stimulate peer-to-peer exchanges which are likely to create cascades in information diffusion and behavioural change [114].

## 4.2. Inclusiveness, valuation, and concerted responsibilities distribution

Although farmers are voluntarily responsive to (expected forthcoming) regulation and to intense policy scrutiny regarding BMPs adoption [16], government intervention alone cannot solve complex issues like cross-boundary nonpoint source pollution. In fact, interviewees (including farmers) recognized the need for regulation and compliance control, but also

stressed the lack of economic valuation and social recognition for voluntary adoption of BMPs. Especially, small-scale farmers reported oppressive economic, political, and social pressure, with limited support or recognition. Inequitable distribution of pressure and responsibilities across sectors was also repeatedly mentioned.

As stated by several farmers-participants, the inclusion and active participation of all relevant stakeholders in the discussion, proposition, co-design and decision process of policy, is necessary in order to take their respective interests and responsibilities into account. Such participatory processes increase the chances to reach consensus among stakeholders, and to develop realistic, equitable, and better calibrated policy mixes. Evidence of collaborative practice and multi-stakeholder partnerships in addressing complex environmental issues is plentiful [115–118]. In the case of Chesapeake Bay watershed, partnerships including representatives from land grant universities, trade associations and businesses, agricultural and environmental government agencies, were established to provide expertise and leadership on policy and programs development. However, this requires available time and human resources from the participating organizations. For example, farmers lack time for off-farm activities, making their inclusion difficult [46]. This could be addressed by valuing or compensating their participation in such initiatives. Administrators (of farmers organizations or cooperatives for example) could play a role in facilitating and incentivizing (farmers) engagement in multi-stakeholders partnership [102].

As inter-farmers interactions were shown to be an effective information channel, incentivising farmers cooperation and collaboration through collective engagement may be an interesting avenue. Recent agricultural policy reforms in Europe have institutionalised such collective schemes [111, 112, 119, 120], which are increasingly recognised as an effective way to coordinate conservation effort at landscape level [103, 109, 110, 112, 113]. In France for instance, agri-environmental schemes can be contracted by collective organizations such as pastoral groups or municipalities [120], while collective incentives (e.g. where farmers are only paid once a certain area coverage is collectively reached) are being implemented [121]. Since 2016, the Netherlands went a step further in institutionalising collective action by choosing to implement agri-environmental schemes through collective applications exclusively (mostly through Environmental Cooperatives) [112, 119, 122]. These novel approaches have potential to rapidly diffuse BMPs, as they may be fostered by peer-to-peer interactions [111]. Such collective schemes can be more effectively diffused by using network characteristics to identify strategic adopters (farmers) and diffusers of the scheme (farmers and non-farmers). This may be achieved by targeting actors who appear as bridging individuals or opinion leaders in the network, or by identifying communities of practice which are more likely to change at the same time (segmentation) [114].

## V. Limitations and future research needs

Throughout this discussion, we have presented potential future policy avenues for increased BMPs implementation based upon both our analyses and existing literature. However, the sample size of this research (number of survey-respondents) is too small to be considered representative of the larger region (Pennsylvania or Chesapeake Bay watershed), or of dairy farmers in general. The method could be replicated with much larger and more diverse samples (for example, by including other types of production and farming systems), in order to determine statistically significant typologies and relations between actor types and messages characteristics. In addition, more qualitative research is needed to understand the common characteristics of farmers who share similar networks. In fact, as we demonstrated that different farmers are linked to different actors, future research should elucidate the criteria of

differentiation. This would allow to investigate and compare the networks of farmer types, which may yield more precise insights for intervention targeting. Similarly, for profit industries could be better characterized, with a differentiation between agricultural inputs suppliers and agricultural outlets. This would allow to determine the role of these different types of industries in the sustainability transition, and the frameworks through which they can be included. Other geographic areas should be targeted, in order to identify commonalities and trends in relational patterns.

We acknowledge that information needs may vary per farmer and per BMP, which might also explain patterns in messages distributions and information flows. This factor of differentiation has not been taken into account in our analysis, and needs to be further investigated in future studies. Moreover, data collection should not be limited to BMP-messages, and should be extended to "farm practices messages", as some actors may strongly influence farmers practices, but may not communicate about BMPs (e.g. dairy cooperatives). These actors could also constitute potential policy targets or entry points, that could have a powerful influence in changing farmers practices.

We have defined strategies to enhance BMP-messages acceptance by farmers, by targeting trusted intermediaries (i.e. message senders who are considered more credible by the farmers). However, it is important to note that, although this may lead a wider range of farmers to consider adopting certain BMPs, it may not lead to actual practice uptake. In fact, in the words of Social Judgement Theory, we have focused on the acceptance "of the credibility and trustworthiness of the intervening actor" [42, 43, 52]. Other factors and layers of acceptance may hinder BMPs adoption despite the message being received by a trusted source, such as the acceptance of the problem definition, or of the perceived consequences or risks associated with the intervention [42, 43, 52].

Farmers preferences, needs, and motivations to engage in conservation action vary greatly, requiring "menus" of conservation measures, where a variety of options are proposed to farmers [123, 124]. Follow-up research could focus on determining types of farmers based on their trusted of information, and on studying relational patterns between types of measures, message source (i.e. the actors through which they are convened to farmers), message acceptance, and actual practice uptake. This would be especially useful to assess if "harder to reach farmers" (e.g. who are unwilling to work with government and research) can be reached via "non-traditional" conservation intermediaries such as supply chain actors.

In addition, more research is needed to understand the subtleties of cascade effects of different policy interventions. As discussed, tailored governance entry points may change BMPs adoption and information exchange by dairy farmers in central Pennsylvania. Further research could both refine policy mixes and provide more content about adoption itself, instead of information exchange specifically. There remains significant uncertainty, not only about governance instruments, but also about the ability for BMPs to positively impact large watershed change [125]. Finally, persistent agri-environmental governance challenges remain rooted in the valuation and accounting system that frame the entire governance structure. Addressing those would require the redesign of standards and valuation frameworks, inclusive of the true-costs of environmental degradation and the true-return of restoration and conservation. Hence, methods and standards to assess, quantify and monitor the degree of liability of actors in environmental deterioration, and their credits/share in natural resources preservation need to be further investigated.

## VI. Conclusion

Water quality governance is a complex and challenging task, complexity which becomes even more apparent through our study. In the context of Chesapeake Bay's nonpoint source

pollution issue, this study has assessed for the first time how information about specific BMPs is exchanged with dairy farmers in a sub-region of the bay, unveiling how the actors network is structured. Therewith we demonstrate how using network data allows to identify (BMP-) message senders whose messages are more likely to be accepted by certain farmers.

Our findings suggest that a variety of tailored policy measures might be optimally combined through specific actor entry points to build better information exchange across stakeholders. For instance, our study confirms the effectiveness of inter-farmer interaction in disseminating farm practices such as BMPs. In addition, certain actors of the for profit sector who are in close interaction with farmers could be viewed as potential conservation intermediaries. This is especially relevant to reach farmers who are unwilling to work with the government, and would allow to alleviate the concentration of pressure and information on farmers.

We have demonstrated that we can increase policy mixes coverage by tailoring governance entry points to specific BMPs and actors, and by building on variability in farmers' networks and exposure. We recommend that policy makers consider the variety of entry points and possible intermediaries that can be involved in encouraging BMPs uptake when designing policy mixes. Thereby the policy mix for BMP diffusion could be broadened, through measures that complement the "menu" of existing incentives, potentially allowing to reach more (and different types of) farmers. In particular, we suggest that:

i. Focus could be put on designing conservation incentives that target supply chain actors as conservation intermediaries, such as cooperatives, processors, or private consultants, instead of directly targeting farmers;

ii. Compliance-control of manure management planning could be conducted by accredited private certifiers;

iii. Policy should focus on encouraging inter-farmers interaction; thus, farmers' mobility, training, knowledge-exchange, and engagement in collaborative multi-stakeholders initiatives should be encouraged via financial or non-pecuniary compensation;

iv. Collective incentives could be considered to better coordinate conservation effort at the landscape or (sub-)watershed scale;

v. All relevant stakeholders (including farmers) should be concerted and included in the discussion, proposition, co-design and decision process of policy, in order to take their respective interests and responsibilities into account.

Further research is needed to better understand if and how information needs differ per BMP, to characterize farmer-type networks, and to determine with which policy instruments and financing schemes these entry points can be optimally combined.

## Supporting information

**S1 File. Survey questionnaires.** I. questionnaire for farmers; II. Questionnaire for organizations.
(DOCX)

**S2 File. Categorization of message types.**
(DOCX)

**S3 File. Guiding questions for semi-structured interviews.**
(DOCX)

**S1 Table. List and description of BMPs.**
(DOCX)

**S2 Table. List and description of the organizations from the socio-centric network.**
(DOCX)

**S3 Table. Codes and themes from interviews content analysis.**
(DOCX)

## Acknowledgments

The authors would like to thank all the stakeholders who participated in this research by voluntarily accepting to share their experience and opinions with us, as well as all the persons who helped facilitating the data collection process.

## Author Contributions

**Conceptualization:** Elsa L. Dingkuhn, Lilian O'Sullivan, Rogier P. O. Schulte, Caitlin A. Grady.

**Data curation:** Elsa L. Dingkuhn.

**Formal analysis:** Elsa L. Dingkuhn.

**Funding acquisition:** Rogier P. O. Schulte, Caitlin A. Grady.

**Investigation:** Elsa L. Dingkuhn.

**Methodology:** Elsa L. Dingkuhn, Lilian O'Sullivan, Rogier P. O. Schulte, Caitlin A. Grady.

**Project administration:** Elsa L. Dingkuhn, Rogier P. O. Schulte, Caitlin A. Grady.

**Resources:** Rogier P. O. Schulte, Caitlin A. Grady.

**Supervision:** Lilian O'Sullivan, Rogier P. O. Schulte, Caitlin A. Grady.

**Validation:** Elsa L. Dingkuhn, Lilian O'Sullivan, Rogier P. O. Schulte, Caitlin A. Grady.

**Visualization:** Elsa L. Dingkuhn, Rogier P. O. Schulte, Caitlin A. Grady.

**Writing – original draft:** Elsa L. Dingkuhn.

**Writing – review & editing:** Elsa L. Dingkuhn, Lilian O'Sullivan, Rogier P. O. Schulte, Caitlin A. Grady.

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
