## [Decision Letter · Decision Letter 0]

17 May 2023

PONE-D-23-07579When unriddling information pathways reveals governance opportunities: A social network analysis of agricultural best management practices in central PennsylvaniaPLOS ONE

Dear Dr. Dingkuhn,

Thank you for submitting your manuscript to PLOS ONE. After careful consideration, we feel that it has merit but does not fully meet PLOS ONE’s publication criteria as it currently stands. Therefore, we invite you to submit a revised version of the manuscript that addresses the points raised during the review process.

We look forward to receiving your revised manuscript.

Kind regards,

Donato Morea, Ph.D.

Academic Editor

PLOS ONE

Journal Requirements:

5. We note that Figure 1 in your submission contain map images which may be copyrighted. All PLOS content is published under the Creative Commons Attribution License (CC BY 4.0), which means that the manuscript, images, and Supporting Information files will be freely available online, and any third party is permitted to access, download, copy, distribute, and use these materials in any way, even commercially, with proper attribution. For these reasons, we cannot publish previously copyrighted maps or satellite images created using proprietary data, such as Google software (Google Maps, Street View, and Earth). For more information, see our copyright guidelines: http://journals.plos.org/plosone/s/licenses-and-copyright.

(1) You may seek permission from the original copyright holder of Figure 1 to publish the content specifically under the CC BY 4.0 license.  

6. Please upload a new copy of Figures 2 and 3 as the detail is not clear. Please follow the link for more information:

https://blogs.plos.org/plos/2019/06/looking-good-tips-for-creating-your-plos-figures-graphics/

https://blogs.plos.org/plos/2019/06/looking-good-tips-for-creating-your-plos-figures-graphics/

Reviewers' comments:

Reviewer's Responses to Questions

**Comments to the Author**

1. Is the manuscript technically sound, and do the data support the conclusions?

Reviewer #1: Yes

Reviewer #2: Partly

2. Has the statistical analysis been performed appropriately and rigorously? 

Reviewer #1: N/A

Reviewer #2: Yes

3. Have the authors made all data underlying the findings in their manuscript fully available?

Reviewer #1: Yes

Reviewer #2: Yes

4. Is the manuscript presented in an intelligible fashion and written in standard English?

Reviewer #1: Yes

Reviewer #2: No

5. Review Comments to the Author

Reviewer #1: The paper presents a social network analysis which was combined with qualitative interviews. The focus of the analysis is to assess possible entry points within the analyzed networks to introduce information on best management practices which are suited to improve water quality to dairy farmers. The analyzed case study is the Chesapeake Bay area in the United States.

A really nice and well written paper. I enjoyed reading it very much!

Below I just list a few points which the authors may consider for a minor revision of their manuscript.

- L58, ‘… yielded insufficient results’: argument would be stronger if backed up with one or two references

- L113, … ‘reject messages from government …’: same here, back up with one or two references

- L162/163: something is missing in this sentence, please check

- Table 2: I wonder why you haven’t included network density as it gives a good account on how well network actors are connected (cf. also L294)

- Table 2 again: typo in the explanation for ‘betweenness centrality’: replace ‘closeness’ with ‘betweenness’.

- L391-399: Here you list which BMPs are mentioned how often, but I wonder if the BMPs are actually equally important to all farmers and if it makes perfect sense that some farmers only mention specific ones. Maybe provide a bit more background info on this aspect?

- L403: replace ‘don’t’ by ‘do not’?

- Section 4.4: backing up your arguments with the statements from the interviews: really nicely done!

- L549, ‘… recent European-level study (2022)’: place reference right here so reader knows which study

- Discussion/Conclusions: You could make more reference to what is going on in other countries to strengthen how your study links to current international developments, e.g. commitment of several European countries to introduce collective agri-environmental schemes where farmers coordinate BMP at landscape level (e.g. Dutch ANLb, German pilots in Saxony-Anhalt and Brandenburg) or group contracts among farmers (e.g. in France).

- Acknowledgements, ‘… all the persons who helped’: maybe acknowledge them by their name? Most people really appreciate when someone says ‘thank you’.

- Fig. 5: explain categories 1, 2, 3, 4 on x-axis

Reviewer #2: Dear authors,

From my point of view, I would recommend writing more about how research can fill the gap in the literature. Add chapter on literature review. I recommend that authors discuss the literature review in a pro-con approach, highlighting the benefits of different approaches in the field while discussing the limitations of the research. I would develop the implications for theory, practice and policy makers in one chapter. The text is understandable and clear, but I would recommend reviewing the English language more. The article would benefit from clearer presentation of how and which theories enrich the paper. The discussion should be placed in a separate chapter and the conclusions should be improved. Check that all tables and graphs have sources and include the most recent ones in the bibliography

I appreciate the effort of the authors and hope they will find these comments helpful in improving their research article in the future.

Best regards

6. PLOS authors have the option to publish the peer review history of their article (what does this mean?). If published, this will include your full peer review and any attached files.

Reviewer #1: No

Reviewer #2: No

---

## [Author Response · Author response to Decision Letter 0]

7 Dec 2023

Dear editor and reviewers,

We are thankful for the insightful feedback that were provided to improve our paper entitled: “When unriddling information pathways reveals governance opportunities: A social network analysis of agricultural best management practices in central Pennsylvania”. Please find attached the answers to all comments, as well as a description of how we adjusted the manuscript accordingly. These are provided in the file "Response to reviewers". Answers to the editor are also provided in the cover letter.

---

## [Decision Letter · Decision Letter 1]

29 Jan 2024

PONE-D-23-07579R1When unriddling information pathways reveals governance opportunities: A social network analysis of agricultural best management practices in central PennsylvaniaPLOS ONE

Dear Dr. Dingkuhn,

Thank you for submitting your manuscript to PLOS ONE. After careful consideration, we feel that it has merit but does not fully meet PLOS ONE’s publication criteria as it currently stands. Therefore, we invite you to submit a revised version of the manuscript that addresses the points raised during the review process.

We look forward to receiving your revised manuscript.

Best regards,

Prof. (Assist.) Donato Morea, Ph.D.

Academic Editor

PLOS ONE

Reviewers' comments:

Reviewer's Responses to Questions

**Comments to the Author**

1. If the authors have adequately addressed your comments raised in a previous round of review and you feel that this manuscript is now acceptable for publication, you may indicate that here to bypass the “Comments to the Author” section, enter your conflict of interest statement in the “Confidential to Editor” section, and submit your "Accept" recommendation.

Reviewer #1: All comments have been addressed

Reviewer #2: (No Response)

2. Is the manuscript technically sound, and do the data support the conclusions?

Reviewer #1: (No Response)

Reviewer #2: Partly

3. Has the statistical analysis been performed appropriately and rigorously? 

Reviewer #1: (No Response)

Reviewer #2: Yes

4. Have the authors made all data underlying the findings in their manuscript fully available?

Reviewer #1: (No Response)

Reviewer #2: Yes

5. Is the manuscript presented in an intelligible fashion and written in standard English?

Reviewer #1: (No Response)

Reviewer #2: Yes

6. Review Comments to the Author

Reviewer #1: (No Response)

Reviewer #2: Dear Authors,

I recommend making the title of the paper shorter. I also advise the authors to explain the methodology in more detail. The discussion part needs to be expanded as well as the conclusions part.

I recommend putting the limitations of the paper in a separate chapter. The references must also include the most recent papers.

Best regards

7. PLOS authors have the option to publish the peer review history of their article (what does this mean?). If published, this will include your full peer review and any attached files.

Reviewer #1: No

Reviewer #2: No

---

## [Author Response · Author response to Decision Letter 1]

15 Mar 2024

Dear Reviewers, 

We hereby respond to the comments to our manuscript originally entitled: “When unriddling information pathways reveals governance opportunities: A social network analysis of agricultural best management practices in central Pennsylvania”. We are thankful for your suggestions and comments which we have addressed or commented in the attached excel file.

We have thoroughly re-examined the manuscript and could not detect any further overlook. We hope to have addressed all concerns, and look forward to hearing from you in due course.

Thank you and best regards

---

## [Decision Letter · Decision Letter 2]

1 May 2024

Navigating agricultural nonpoint source pollution governance: A social network analysis of best management practices in central Pennsylvania

PONE-D-23-07579R2

Dear Dr. Dingkuhn,

We’re pleased to inform you that your manuscript has been judged scientifically suitable for publication and will be formally accepted for publication once it meets all outstanding technical requirements.

Best regards,

Prof. (Assoc.) Donato Morea, Ph.D.

Academic Editor

PLOS ONE

Reviewers' comments:

Reviewer's Responses to Questions

**Comments to the Author**

1. If the authors have adequately addressed your comments raised in a previous round of review and you feel that this manuscript is now acceptable for publication, you may indicate that here to bypass the “Comments to the Author” section, enter your conflict of interest statement in the “Confidential to Editor” section, and submit your "Accept" recommendation.

Reviewer #2: All comments have been addressed

2. Is the manuscript technically sound, and do the data support the conclusions?

Reviewer #2: Yes

3. Has the statistical analysis been performed appropriately and rigorously? 

Reviewer #2: Yes

4. Have the authors made all data underlying the findings in their manuscript fully available?

Reviewer #2: Yes

5. Is the manuscript presented in an intelligible fashion and written in standard English?

Reviewer #2: Yes

6. Review Comments to the Author

Reviewer #2: Dear authors you responded adequately to the additions.

I hope additions as suggestions to improve your manuscript.

Best regards

7. PLOS authors have the option to publish the peer review history of their article (what does this mean?). If published, this will include your full peer review and any attached files.

Reviewer #2: No

---

## [Editor Report · Acceptance letter]

9 May 2024

PONE-D-23-07579R2 

PLOS ONE

Dear Dr. Dingkuhn, 

I'm pleased to inform you that your manuscript has been deemed suitable for publication in PLOS ONE. Congratulations! Your manuscript is now being handed over to our production team.

Kind regards, 

on behalf of

Professor (Associate) Donato Morea 

Academic Editor

PLOS ONE